# Stakeholders' Views on Responsible Assessments of Assistive Technologies through an Ethical HTA Matrix

**Erik Thorstensen** 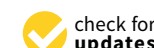

Work Research Institute, OsloMet—Oslo Metropolitan Univeristy, NO-0130 Oslo, Norway; erikth@oslomet.no

**Abstract:** Assessments of novel assistive technologies for use in home-based services has been documented to be performed in a variety of ways and often with a rather narrow focus on safety and effect or effectiveness. In order better to understand the place for wider forms of assessments of assistive technologies, the current study presents a combination of the Ethical Matrix and the Socratic approach for assessment of health technologies—the Ethical HTA Matrix. This matrix was filled with content based on a case of a GPS localization system, which was validated by stakeholders. In a next step, central decision-makers in assistive technologies and stakeholders were interviewed concerning their views on this methodology. Mainly, the matrix was seen as very comprehensive, but too detailed with an abundance of information. Nevertheless, some informants suggested concrete uses of the matrix in their organizations. Some understood the matrix more as an epistemic tool aiming at providing an overview of the state of knowledge, while others identified a normative potential in the matrix that could be implemented in health innovation processes for the home-based services, in particular when discussing novel solutions and working methods with health professionals and care workers.

**Keywords:** responsible research and innovation; assistive technologies; ethical matrix; health technology assessment; GPS tracking; ethics

## 1. Introduction

Communities, municipalities, counties, regions, countries, and supranational structures all attempt at successful integration of assistive technologies into care for persons with disabilities and older adults with needs for assistance. At the same time, a range of disciplines from medicine, nursing, and manual therapy on the one side to engineering and informatics on the other side have conducted and will continue to conduct research, development and innovation in assistive technologies. In addition, there is a substantial contribution from the social sciences and the humanities to the social, ethical, cultural, and legal dimensions of ageing, technology and assistive technologies. The main purpose of research and innovation in assistive technologies is to increase the number of people who might live in their own homes for a longer time. This purpose has two underlying rationales: (1) People prefer to live in their own homes rather than in some form of institution; and (2) Societies might avoid significant increases in costs when or if fewer persons live in some form of institution.[1] These two rationales have a weighty normative content. The first point to the good life and the second refer to the survival of the political and geographical entities mentioned above through a control over spending. I believe that

---

[1] Both of these underlying rationales might be questioned. According to Oswald, et al. [1] and Wahl, et al. [2] the view that belonging is the main factor for quality of life needs to be complimented with an understanding that housing-related agency is just as central for maintenance and construction of identity. For the second rationale, Okunade and Murthy [3] and Fineberg [4] have shown how technological systems are drivers of cost in the healthcare system.

it is of importance to underline that the introduction of ICT-based solutions as measures to provide prolonged residency has existed since the beginning of the 1990s. Furthermore, the uptake of such solutions could best be characterized as hesitant [5].

In this contribution, I will present, examine and discuss one possible approach to assess an assistive technology in an integrated manner. My objective is to provide a tool for decision-makers responsible for acquiring assistive technologies and for developers of assistive technologies that can serve as a structuring device for assessing costs and benefits at the same level as the wider social, ethical, cultural, and legal aspects. Through a series of interviews with central stakeholders and persons whose daily work is to be the mediator between the political ambitions in assistive technologies and the concrete implementation in the services, I aim to not only find their opinions on the tool, but also aim at finding possible places for the approach in the municipal health innovation processes, which is a field with relatively few studies [6].

The normative context of Responsible Research and Innovation (RRI) provides the frame for the current research. Here, the importance of integrated assessments has been underlined by von Schomberg [7], while a related care for the future impact of new and emerging technologies has been voiced by Owen, et al. [8]. Relevant to my argument is the demonstration by Fitzgerald and Adam [9] of the fragmentary decision-making landscape in Europe for assistive technologies, the lax and seemingly arbitrary approaches to testing of assistive technologies [10,11], and the recent report from the European Parliamentary Research Service Scientific Foresight Unit (STOA) voicing concerns over lack of proper assessments of assistive technologies classified as consumer technologies, which constitute the large majority of the marketed products [12]. Hopefully, my contribution might serve as a first brick in a bridge between the hasty, short-spanned, and profit-based logic in industry and the careful, person-centered, and budget-based logics of the health sector, as proposed by Demers-Payette, et al. [13], through the use of similar assessment instruments.

In this article, I introduce and discuss the concept of responsible assessments based on recent work in RRI and assessment methodology of new and emerging health technologies [14–21] before presenting a methodology, which is a combination of the Socratic approach to ethical analysis of health technologies and the Ethical Matrix [22–25]. I apply this methodology, the Ethical HTA Matrix, on an assistive technology currently on the market in several towns and municipalities in Norway and with increasing use, and present the results from the analysis to decision-makers in assistive technologies. Bruijnis, et al. [26] have already applied the Ethical Matrix in the context of RRI and they see it as contributing to the realization of RRI as presented by Owen, Stilgoe, Macnaghten, Gorman, Fisher and Guston [8], with a caveat for responsiveness. What is my main theme in this article is the feedback from the decision-makers on the Ethical HTA Matrix. Based on these presentations and the ensuing discussions, I make recommendations for future changes and contribute with more general comments as to the possible places, desired formats and content for such assessment.

## 2. Materials and Methods

The concept of responsible assessments is based on the thinking and practices of RRI. The current approach to RRI has attempted to remain open as to what it signifies to perform research according to RRI. The explicit intent in this paper is to bring the voices and concerns of users and stakeholders closer to the political processes of producing and acquiring assistive technologies with the aim of shaping innovation processes through a normative model. Here, I aim at highlighting other societal processes and concerns that might have an impact on the usefulness, but also on the acceptability of the proposed assistive technology. A central theme in the discussion is precisely at what time in the policy process, on the one hand, and in the innovation processes, on the other hand, the Ethical HTA Matrix might be beneficial for the respondents. Such a link between the political processes and stakeholder events has been described as a "strong RRI approach" by Coenen and Grunwald [27]. Underlying this search, is an interest in how research and innovation in assistive technologies might be

governed in a way that makes the products available to users with special and specific needs both more acceptable and desirable [28].

## 2.1. Background

In RRI, the quest for the right impacts is a central theme [7,29,30]. Such right impacts are notoriously difficult to define precisely, but they should consist of the realization of social values and contribute to the preservation of public goods and maybe even enrich the latter. In the European context, von Schomberg [30] has suggested to connect the values to those of the Treaty of the European Union, while the public goods relate to the protection of the environment, human health, and sustainability. As indicated in the introduction, there is a large policy pull to realize assistive technologies, and simultaneously, there is a strong technology push to introduce these in home-based services. von Schomberg [7] has proposed that assessments might be used in RRI as a moderator between the search for the right impacts and the push and pull forces. Since assistive technologies in the homes to a large extent fall under the regulation for consumer goods rather than for medical devices, there are no set procedures outside of technical functionality and absence of harmful components regulating the products. This situation indicates that there is a large political space for discretion and a variety of possible methods for assessments. The mentioned STOA report suggests narrowing this gap through a more nuanced classification system [12]. However, the question remains how to moderate and how to narrow these gaps. In this respect, Forsberg, Quaglio, O'Kane, Karapiperis, Van Woensel and Arnaldi [20] have introduced the concept of responsible assessments for assessment approaches that meet the normative ambitions of RRI as described by Wickson and Forsberg. They present a view where the responsibility of research and innovation can be described by its ability to

1.  address significant societal needs and challenges;
2.  engage a range of stakeholders for the purposes of mutual learning;
3.  anticipate potential problems, identify available alternatives, and reflect on underlying values; and
4.  respond, act and adapt according to 1–3 [31].[2]

My scope in this article is to provide an assessment for responsibility, which refers to "the support that the assessment apparatus can provide for responsible development and governance of science and technologies" [20]. I present a method that I investigate whether could be used in order to align governance with Wickson and Forsberg's 1–4 above. However, in order for an assessment to be fully responsible, the process of creating the substantive content of an assessment methodology in this field has to be conducted with responsibility, i.e., in a manner that meets the four requirements and that might be applied to an existing product.

In order to find what the right impacts might be in the case of an assistive technology in a Norwegian setting, a transdisciplinary groups has used fieldwork, user workshops and literature studies to approach how and why users and different stakeholders value assistive technologies [16,32,33]. A brief summary of the main areas of concerns and that any assessment should address are the good life, risks and benefits in use; risks and benefits before use; the distribution of risks and benefits; and the distribution of responsibilities and training, in addition to opening up on the changing nature of future impacts of assistive technologies. These findings informed a survey and evaluation of different assessment methodologies for new and emerging technologies [17]. A central theme here was further the applicability of the methodologies in the health domain and more specifically on assistive technologies.

The proposed methodology, the Ethical HTA Matrix, is based on a combination of the Ethical Matrix [24,25] and the Socratic approach for ethical analysis of health technologies [22,23]. These two

---

[2] This description encompasses what Owen, Stilgoe, Macnaghten, Gorman, Fisher and Guston [8] have presented as Responsible Innovation, namely anticipation, inclusion, reflexivity, and responsiveness, but adds what von Schomberg [30] has described as "grand challenges".

were primarily selected to address a wide range of concerns that an RRI assessment should include for assistive technologies [17], and in addition they have been widely used to analyze ethical and social impacts of novel technologies [10,14,19,21,22,24,26,34–50]. The reason for combining these two is that the Ethical Matrix can best be characterized as a general-purpose tool for practicing applied ethics, whereas the Socratic approach is highly specified towards the health field. The Ethical Matrix serves well to illustrate how the same technology or product can affect different stakeholders in different ways, which is a less visible feature in the Socratic approach. The Socratic approach then provides the concrete questions whereas the Ethical Matrix gives the structure to discuss how the issues influence the relevant stakeholders.

*2.2. Methods*

The object selected to make a test case for the Ethical HTA Matrix (hereafter, the Matrix) is a GPS (Global Positioning System) localization tool that includes next-of-kin as primary responders, but with the public health system as a secondary responder. This GPS system is a commercial system currently being used by a number of municipalities in Norway. The GPS system is mainly used for persons with dementia or other sever cognitive conditions. The decision to implement a GPS system has to take place with the consent of the person being tracked (hereafter the user). If the user lacks capacity to consent, a legal decision is needed to implement GPS tracking for the user. The GPS system consists of three basic components: a tracking device to be carried by the user so that she or he is possible to locate, a steering system keeping track of all users and relating users to the ones responsible to locate the tracking device (and the person), and a smart phone application to be installed at the phones of those with a legal right to participate in tracking and with a corresponding obligation to act as a first responder. These latter are often family members and will be referred to as "next of kin". The tracking device might activate an alarm if the device moves outside of a predetermined geographical area (a so-called geofence) and if the user triggers the alarm physically. It further has the possibility of two-ways communication since it is based on both GSM (Global System for Mobile Communications) and GPS. If the first relevant next of kin does not respond to an alarm, the alarm can be routed to a different next of kin, and in the last instance to the health services. Hereafter, I will refer to this system as "the GPS system".

What is of interest here is how the different values in the Matrix, as specified by the Socratic approach, are affected by the introduction of the GPS system: Will a primary user experience that she has more freedom to move? How important is it for her to move? How sure are we that our knowledge is correct? And: How much money will the health system save? How important is money-saving for the health system? How sure are we of the savings? These are the types of questions the Matrix will attempt to display.

2.2.1. Completing the Matrix

The first step is to identify the stakeholders affected by the introduction of the GPS system and place these in the matrix structure (cf. Table 1).

Since the Socratic approach seems both relevant and complete concerning novel technologies in health, the next step is to identify which questions from the Socratic approach could be placed where in the matrix (see Appendix A for completed Matrix with questions from the Socratic approach).

The Matrix might be used in several different ways, but there are two main approaches. The first consists in finding all the values for the different stakeholders together with them or with their representatives. A different approach is based on desk research on what the research literature and other policy-relevant documents say about these values [35]. In this article, I have chosen the second approach. The key to applying the Matrix consists of two main phases: (1) document the relevant values at stake for the different stakeholders, and (2) validate the impact of the technology for the relevant values with the stakeholders. The force of these impacts can be described in different manners, but often one applies a scale that might be a 2-point (important, very important) or along a range

−5 to +5. When having completed a Matrix, the full material is presented to a decision-making or policy-making body either in order to reach a decision or to inform on a subject matter.

**Table 1.** Stakeholders[3] affected by the GPS system.

|  | Well-Being | Dignity | Fairness |
|---|---|---|---|
| Primary user |  |  |  |
| Health professionals |  |  |  |
| Health delivery system |  |  |  |
| Technology providers |  |  |  |
| Next-of-kin |  |  |  |
| Climate |  |  |  |
| Ecology |  |  |  |

One central underlying idea in this assessment experiment has been to place the values "effect" and "cost", which are central to decision-makers as well as to HTA, under the heading "Well-being" in the Matrix. The rationale for this choice is that these values, which often are given a very prominent place in the decision-making process, should be placed on the same level as other values (see Refolo, et al. [51]).

### 2.2.2. Validation of the Matrix

A central aspect in the work with matrices as methods is to obtain some form of validation from the relevant stakeholders. My field of investigation here does not center on how the different stakeholder groups value, see or understand a GPS system, but rather how the decision-makers involved with assistive technologies perceive the Matrix as an instrument for making decisions. Consequently, the validation phase has been somewhat briefer than the full participation through workshops to weigh the different values or consequences [34,35]. Maybe the most important stakeholder category, the end users, are people with dementia. For these respondents, I employed a brief questionnaire developed in cooperation with the dementia center that they attend daily. Drawing on recommendations from Kennedy and Ter Meulen [52], I conducted individual interviews in familiar settings for a duration of maximum 20 min where we had the GPS tracker between us and talking about different situations where the GPS tracker figured. From these five interviews, there were some points where the respondents disagreed and some points where they all agreed—see Appendix B, The validated Matrix.

For the remaining stakeholder groups, I developed questionnaires distributed by email by a local municipality to next of kin, health professionals, alarm center employees, and health services officials. In this case, I similarly settled for a smaller number of respondents in order not to place too large a burden on the time of health workers in the municipality.

### 2.2.3. Interviews

In order to obtain views and opinions on how the Matrix is perceived among those working daily with making decisions concerning assistive technologies, I made arrangements for interviews with two representatives from civil society organizations, two technology developers and nine representatives from health decision-makers or advisors working for or with mandate from towns, municipalities and government in March 2019. They were selected based on experience with assistive technologies from their respective perspectives. The interviews lasted between one hour and two hours and were recorded with written consent from all the respondents. Prior to the interviews, I sent the Matrix (Appendix B) to all interviewees with an explanation about the project and methodology.

---

[3]    The last two stakeholders, the climate and ecology, are stakeholders to the extent that these are entities affected by technology. In traditional applications of the Ethical Matrix in biotechnology, the ecosystems play a central part. Pacifico Silva, Lehoux, Miller and Denis [18] argue that RRI in health need to take into account how the health system affects the climate.

In each interview, we were between one and three persons. I began the session through a brief presentation of myself and the Assisted Living project, and asked them how they understood the acceptance, uptake and use of assistive technologies in their context. Mostly, this served as a method to start the conversation and make the interviewees talk about their experiences. These experiences are relevant because the methods for product assessment need to also be relevant within their local challenges and for their professional tasks. This interview strategy is what Spradly calls task-related grand tour questions [53].

Interviews with experts in the field, such as my informants, presuppose an active interviewer well-orientated in the field pursuing an open approach to the theme based on general topics [54]. To introduce the experts to give their own narratives, such as grand tour questions, is recommended in order to provide insight into more unreflective or tacit knowledge dimensions [54].

Following this overview, I turned the attention to the assessment of the GPS system as presented in Appendix B in order to establish a common ground for the discussion of the Ethical HTA Matrix.

I further used material from earlier interviews and suggestions and positions from these towards the end of the interview in order to hear others' opinions and judgments of the previous informants' views. This methodological position places the researcher more as an active part in the creation of knowledge and spreads and disseminates thoughts and insights between the informants. Furthermore, given the wide discussion in the introducing themes, I wished to investigate where in the innovation process an RRI method could provide additional value, thus reflecting the debate on the place of responsibility in innovation processes [55]. As Reijers, Wright, Brey, Weber, Rodrigues, O'Sullivan and Gordijn [15] point out, there are several possible points for normative intervention in a decision-making process which also affects the choice of method.

*2.3. Stakeholders' Values*

Since I have selected to start with desk research and validate the impacts with stakeholders, the first step was to identify relevant literature. Fortunately, there have been a range of development and research projects conducted in Norway on different aspects of GPS tracking technology as an assistive technology in the period from 2012 to 2018. For the purposes of developing and testing the Ethical HTA Matrix, I took these reports as a point of departure and worked backwards in the approach called "snowballing", i.e., searching for relevant references in the reports [56]. This limitation to Norway might be questioned, but it simultaneously avoids the problems inherent in comparing between national health systems and, furthermore, several of the studies from the snowballing cover a wider spectrum of contexts.

2.3.1. Primary Users

Concerning the effects for the primary users, GPS systems seem to increase mobility and daily activity, but only for users who are already active [57–59]. Users stop benefitting from GPS systems when they are no longer capable of going outdoors [60]. GPS systems further contribute to faster location and retrieval of the users when they trigger an alarm or they are missing [59]. When it comes to the possibility to live at home for a longer period of time, studies indicate that a GPS system can be beneficial [61], but for a minority of the users (15–20%) and for a period of 3 to 8 months on average [62,63].

In general, GPS systems increase freedom of movement and experience of freedom [61,64] as well as increase interpersonal contact and maintain friendships [60]. In addition, a GPS system including a geofence system would be even more adaptable to individual preferences and to avoidance of neighborhood dangers than a simple tracking system [60]. The fit between user and device might be problematic due to the illness or to malfunctioning of the GPS system [59,62,65,66].

One connected risk element is that GPS systems might create a false sense of security when users lose the ability to dress for cold or rainy weather [64].

A remedy to some of the risks is to invite users to start using GPS tracking at an earlier stage in order for the users to master their devices [63] and open up for learning and experimentation [66]. This solution, however, might create opposition and will place temporarily increased stress on the health system in addition to creating possible legal challenges as the current use is based on health needs [67].

Some of the trackers might be experienced as stigmatizing [57,63]. The issue of surveillance and privacy is challenging when it comes to GPS tracking for users with cognitive impairments [68]. In addition, the users' privacy is affected and even more so if the system also included a geofence function [69,70]. Even though the large majority of the users do not feel surveyed [63], some report tracking as affecting their privacy [61,69]. To complicate things further, users might feel surveyed even if they do not report it [71], and a majority of the users report that they primarily use GPS for the sake of the peace of mind of others [57,61,64]. It is consequently a need to secure that consent is non-coerced [69]. As the alternative to GPS tracking in some cases is drugs or physical restraint [59,61,62] or being admitted to a health institution or assisted living facility [71], voluntary consent becomes even more difficult to assess. Most users report increased independence and higher quality of life [64] while not experiencing a loss of authority over their own lives [57].

Concerning matters of fairness, it seems that the introduction of GPS systems favors those already co-residing with someone [63], and that a GPS system with next-of-kin as first responders would favor those with next-of-kin nearby. Likewise, it is reported as easier to give training to and maintain the use of GPS for persons living with others than those living alone [64]. Specific efforts are needed to target persons residing alone [60,63], and these are the ones who would benefit the most from using GPS tracking [64]. As should be clear, GPS systems might primarily benefit those capable of walking around on their own [72].

### 2.3.2. Health Professionals

The main value of GPS systems for health professionals is a workday with more focus on health-related tasks. This effect has its causes in less searches for people because they are missing or because they are not at home when the health professionals have an appointment to see them [61,72,73], but also in fewer alarms interrupting daily planned routines [74]. An increased feeling of safety among health professionals is widely reported [61,62,64].

The maintenance and charging of trackers, as well as reminding users to carry them with a tracker, are reported as novel tasks and an increase in workload [63,64,73]. An additional task is to assess users in order to understand if they might benefit from GPS [64], as well as monitoring and assessing these benefits [63], and how to include relatives in healthcare [69]. Internally, among health professionals, GPS systems could create the need for new workflows and novel forms of cooperation [73,74].

When it comes to health professionals' performance in their work, such remote-sensing technologies deprive health professionals from continuous contact and might impede observations of deteriorating condition [73] and novel challenges arise as the health professionals know where a patient is, but not what she or he does [61].

Furthermore, the involvement of relatives and the facilitation of contact with relatives might be reducing the need for care services [57]. The involvement of relatives might introduce conflicts over the suitability of a tracking device for a person between health professionals and next-of-kin [69]. In terms or fairness or justice, the health professionals might find themselves in novel legal territory if GPSs are implemented earlier to persons without direct health needs but who are at risk of developing such a need [67]. From the literature, it seems that the physical aspect of care is delegated to the domain of female informal caregivers and low-paid females [75].

### 2.3.3. Alarm Center

For the staff at the alarm center and the alarm center as an organization, the ability quickly to find people outdoors without extensive searches is the main effect [61,72,73]. The main costs relate to upscaling from a limited number of users and to integration between other possible alarm systems

of databases [72]. However, there are some uncertainties relating to the effect. In the cases where one seeks to talk to the users to assess the seriousness of an alarm, the sound quality is a critical factor [72]. A puzzling factor here is the large number of false alarms. One report estimates that 90% are not real alarms [63]. However, studies point to such false alarms as a positive social experience for the users [75] Related to the issue of upscaling mentioned above, is also the challenge of recruiting competent personnel [72], and related to the user interface, one study mentions that the language and expressions used there might influence the personnel's perception of the users [76].

Pertaining to fairness, one study indicates that more time and resources are dedicated to persons using mobile alarms than immobile alarms [72].

### 2.3.4. Health System

The two main purposes of a GPS system for the health system is to provide better quality of life for the users and to reduce the costs of healthcare services. In addition, there is the specific RRI concern of *frugality*, i.e.,: "Does the technology deliver greater value to more people using fewer resources? Does the technology presuppose a larger technological infrastructure?" (see Pacifico Silva, Lehoux, Miller and Denis [18]). In one way, the distributed nature of a GPS system with next-of-kin as first responders is lessening the pressure on health services, but if the infrastructure becomes more complex due to the involvement of new parties, then the criterion of frugality is not met.

The health delivery should become more effective and the staff less stressed with more active users [72], and it is not likely that the number of health professionals will be reduced through a GPS system [61]. In addition, the health service delivery chain and work practices would need at least some reorganization for large-scale implementation [57,63,64,77]. A central feature of a GPS system is to prevent persons going missing. Estimates show that half of those with dementia missing for more than 24 h die or are seriously injured, and the cost of a missing person search was estimated to be in 2012 between £1325 and £2415 in the UK [78].

More active and mobile users should overall give less stress on the services [57] and the professionals would spend more time on health delivery [63]. However, the most significant uncertainties regarding costs is the lack of systematic studies and consistent methodology on the one side [60,79,80],[4] and insight into which patients that will benefit from GPS systems on the other side [64,83], as mentioned above under Primary users.

When it comes to fairness considerations for the health system, one central question is whether the benefits from a GPS system with next of kin as primary responders are distributed fairly. Such a system benefits those who are already active and who have next-of-kin available [72]. The legal situation is further unclear whether or not users should pay, and this might be practiced differently between cities or municipalities [63]. There is further a need for a legal framework that can cope with the challenges of making next-of-kin responsible for health services and safety [84].

### 2.3.5. Next of Kin

Next of kin are already first responders to GPS, but with the introduction of a phone-based application one is no longer dependent on living together with someone in order to maintain this role [57,72]. Overall, next-of-kin report experiencing more safety for their loved ones [72], increased freedom for themselves and the users [60,62,64], and a general peace of mind [69]. It seems that the role of first responders change the prioritization of next-of-kin. The prioritization of family caregivers change from an emphasis on safety when they are responsible to an emphasis on autonomy when the professional caregivers are responsible [69]. In addition, there is a possibility that the introduction of

---

[4]  This lack is widespread in the whole sector of assistive technologies for prolonged residency at home. Franck, et al. [81] found very few validated studies for long-term effects. As demonstrated by Steffensen [82], cost-utility analyses in small municipalities are very sensitive to small fluctuations in staff.

next-of-kin might cause disagreement between users, next-of-kin and health professionals regarding the suitability of both the decision to track and of the tracking device [69].

### 2.3.6. Technology Developers

The studies included here did not pay much attention to the values of the technology developer. In general, I would state that in assistive technologies, the technology providers or developers remain rather understudied.[5] In order to understand better what the values of a technology company in this field are, I approached the producer of the GPS system for an interview. Seen through the dimensions of the Matrix, the ability to make income to the firm and avoid huge expenses with uncertain gain are the main factors under well-being. The more interesting findings I have categorized under dignity and fairness. With regards to dignity, I had the impression that recognition of providing a valuable service is central, but, in addition, they felt very distant from the goods that they deliver to the patients or to the users. The public calls for tender with a too-high degree of technical specificity made it both difficult and tiresome to respond to since the space for novelty is very limited and because it violates a "natural" division of labor between the health services who should be experts on health needs and the technology developers who should know best how to technically configure such a solution.

In the considerations over fairness, too-detailed calls for tender were not considered fair since they more or less then gave preference to one firm over others. Consequently, the ability to compete on fair and even terms seems important and is connected to a view of professional pride in how best to solve the task. The public procurer should also be careful in applying the total of its purchasing power since the public is the single most important possible purchaser in this domain in Norway.

### 2.3.7. Climate and Ecology

Even though the public are the dominant purchasers of assistive technologies in Norway, there is little evidence—or more precisely no evidence—that municipalities or the state considers ecological impacts. According to the Norwegian government, both the environment and climate figures as goals to be realized also by the health sector [87], but the responsibility is placed on hospitals and regions and not municipalities who are responsible for assistive technologies. Other national strategies [88,89], recommendations [62,65], and procurement guides [77] for assistive technologies do not reflect the government's ambition of a more climate friendly or ecofriendly healthcare. A recent Canadian study found that 27 out of 92 requests for proposal for medical technologies included environmental concerns as evaluation criteria [90]. Seen from the perspective of Responsible Innovation in Health [18], an absence of use of monopoly power would qualify as irresponsible especially since there is a large amount of studies documenting the negative life cycle impacts of mobile technologies [91].

### 2.4. A Value HTA Matrix

Based on the concerns identified for the different stakeholders, I arranged the central values at stake into a Matrix (see Appendix B). These values are then expressions of how different stakeholders might be affected through the implementation of a GPS-based tracking system with next of kin involvement. One challenge I experienced in reviewing the literature, was whether to differentiate graphically between the uncertainties and the variations of the effects in the Matrix. Take for example, the finding that almost 50% of the users seemed able to benefit up to 1 year from a GPS tracker when it comes to independent living. It seemed to me untenable to present the value of "Living at home for a prolonged time" with the same certainty as "Being found when lost". Here, I chose to identify the level of certainty with a range of colors from red (uncertain hypothesis) over yellow (some level over certainty) to green (documented consensus). Through this choice, I conflate two

---

[5] This is in contrast to other aspects of health technologies. See Brown and Webster [85] and Lehoux [86] for extensive studies and overviews into technology production and policy processes of medical and health technologies.

different elements: variation and uncertainty. However, at the same time, I assume that variability and uncertainty both relate to the expected effect [92]. Consequently, for the persons making a decision with respect to a desired function, the inside of the black box of the GPS-based tracking system with next of kin involvement might not matter too much, but this assumption will be investigated during the encounters with the decision-makers.

When filling in a Matrix, it is customary to insert what are the likely positive or negative consequences for the different stakeholders following the introduction of the technology [36]. However, when working with the findings from the literature, it became apparent to me that there are a range of important prerequisites that need to be in place or that might even affect the likelihood for the realization of one or several values.[6] Returning to the example of "Living at home for a prolonged time", the literature indicates that one important condition for this value to be realized is that those receiving a GPS tracker are properly diagnosed both with respect to their internal and external capabilities [60,94]. Knowledge of these conditions are important for decision-makers and consequently they should be presented together with the affected values.. At first, I considered presenting two different versions to the informants; one with values and consequences and one with values and necessary conditions. However, I rather listed the conditions calling them "critical factors" and discussed the format with the informants.

## 3. Results

In the current section, the results are grouped according to general comments, views on the applicability of the Matrix on the case of GPS localization, views on the approach chosen to apply colors and fonts through literature review and external validation, and finally suggestions for where in the municipal innovation processes or working flows such a Matrix could be used. The version of the Ethical HTA Matrix used in the interviews is presented in Appendix B translated to English from Norwegian by the author. In order to provide some context, I start the presentation with some of the general challenges and continue with what the informants presented as some of the main changes in the innovation landscape of assistive technologies.

### 3.1. Innovating Assistive Technologies

Several of the informants had been working with assistive technologies either as producers or procurers—or simply as interested parties—since 2011. This year marks the start of the current period of interest in assistive technologies, which started with a White paper [95]. All parties told a story of testing technical solutions in rather limited scope with little or no attention to the integration of these assistive technologies into the services. Largely this period was characterized by a technology push with small firms on the one side and enthusiasts in the services on the other side and politicians with ambitious goals above. A change started to manifest itself around 2015 with sharper focus on the services and lower confidence that the technologies by themselves would revolutionize health. Such a change presupposes increased competences in implementation and workplace innovation. With the creation of the Directorate for e-health in 2016, the whole field has become more professionalized on the one side and now follows centralized guidelines for procurement, but on the other side, this centralization has been advantageous for larger companies that can use smaller technology developers as subcontractors delivering standardized solutions that municipalities implement. Several of the informants saw clear benefits of this change, but others expressed concern that it might stifle technological innovation when smaller firms just become suppliers rather than developers. In the larger municipalities and towns, informants told of changes in workflows, routines and organization as a means to achieve both

---

[6]  One of the criticism levelled against some bioethical approaches has been the uncritical acceptance of optimistic technology futures [93]. Through underlining the necessity of specific factors and conditions necessary for realizing the intended goods, I believe the current approach meets this criticism.

implementation of assistive technologies and improved care. In smaller municipalities, there was a struggle to prioritize organizational change in parallel with technology implementation, and thus it is difficult to achieve innovation as the working routines remained the same. As some informants told me, in smaller municipalities, everything tends to depend upon a few central persons for securing the routines and if these are absent for 1 week, the routines return to the old ways.

### 3.2. An applicable Matrix?

I introduced the Matrix to the participants through the case of the GPS system. First, they were asked if they had any initial reactions. The clearest and unequivocal response to the Matrix was that the suggested approach did not provide an easily accessible overview of the GPS system because of the high level of specifications. The additional outcomes of the discussions might be separated along two strands: One pragmatic strand where the display of information and the applied case is considered, and a different strand where the Matrix is seen as an RRI-tool (see Wickson and Forsberg above). I will commence with the latter strand.

When it comes to the how the Ethical HTA Matrix is related to RRI, several of the respondents gave the impression that it opened up for reflection on problems, mutual learning and value analysis. What characterized all meetings with municipal decision-makers was extensive discussions regarding the red and orange colors in the well-being column for the stakeholder category Health system. They all expressed that the economic effects of introducing assistive technologies are real and substantial, but that there are methodological difficulties in asserting or quantifying these effects due to expected gains in other sectors in addition to those in the home-based services. Most interviewees agreed that even if this latter factor is taken into account, systematic knowledge concerning economic gains is scarce.

One could note an ambivalence regarding care workers and health professionals' roles in innovation processes. Some informants saw their primary role as adapting to the needs of the health system and the users, whereas others regarded their contribution as constitutive of any municipal innovation process. A third view was to rethink the whole organization of the services based on the introduction of assistive technologies. I would venture that there might be different models for innovation underlying these conceptualizations of health professionals as stakeholders. The first approach connotes a view on the care workers as in a principal/agent relation [96], whereas the second moves towards the opposite outlier where one departs from the local experiences and professional values of care workers and health professionals in order to arrive at innovations [97]. The third approach seems managerial with a solid belief in planning and structuring to facilitate for innovations [98].

There is a conflict in the literature between a view on assessments as in need of proper resources and a view suggesting that assessment frameworks should be cost-minimizing [99,100]. This tension could also be found among the interviewees with some highlighting that complexity is a value in itself and one needs to dedicate the proper resources, and others who expressed that municipalities need facile and simple procedures throughout an assessment or procurement process. A different line of comment addressed that what is needed is a closer examination of the localized understanding and a systematization of existing practical knowledge. This line of thinking seems to suggest an additional place for bottom-up approaches or ethnographic studies. The latter might be difficult to reconcile with the procedural thinking in the Ethical HTA Matrix since it emphasizes rich context and local cultures [101] whereas the former approach has been used in several instances [24].

Contrary to the usual practice of using an Ethical Matrix, I have chosen to display what I refer to as Critical factors for the realization of potential values. This dimension addresses the non-use of assistive technologies and barriers to use of assistive technologies (see e.g., Scherer [102]) and the point raised by Hofmann, Droste, Oortwijn, Cleemput and Sacchini [22] regarding the morally relevant challenges of assessing *ex ante*. They expressed that a link between the different levels of critical factors and the realizations of the relevant values is a valuable contribution.

Regarding mutual learning, nearly all the respondents recognized the different stakeholder groups and saw them as relevant with the exception of the climate/ecosystem. The exception to this was a

position where only the health system and the user ought to count. The climate and the ecosystem were not in themselves considered irrelevant, but more out of scope of what decision-makers in assistive technologies might influence. Decisions on these matters are taken at a political level and implemented as general procurement rules, and form part of the calls for tender.

Several informants raised the issue of uncertainty and knowledge as central. However, there were differences in opinion on what counted as knowledge. In the matrix, I relied on published sources whereas some informants added that they had much knowledge—and this lead to a discussion on what counts as knowledge in assessments. Some municipalities conduct a range of studies by themselves, and there is a significant amount of information-sharing concerning assistive technologies between municipalities.

A related discussion to the status of knowledge were the views on who should count as the most relevant stakeholders. In this regard, the technology developers expressed interests in all stakeholder groups but with a clear orientation towards users and next of kin. On the opposite side of the spectrum, an informant from the municipalities said regarding the technology developers, "Finally, they are experiencing our power" whereas a different municipal interviewee saw co-production of services with the technology developers as crucial in fitting solutions to the actual context. The technology developers' orientation towards users and next of kin seems to make sense since these categories constitute their end-target group and it fits their rationale. However, the divergent orientations towards technology developers as a stakeholder group indicates either that large parts of the customization of the solutions takes place in the municipalities or that these municipalities are able to make very specific orders from the technology providers. Regardless which interpretation is correct, there is a peril of little feedback from the municipalities to the technology firms. However, in the opposite case with a large degree of cooperation, this feedback would seem to be secured. These differences were raised directly in the interview with one view held that the technology firms for too long have played a strong part in the implementation of assistive technologies. Whereas a different view was to see the long relation with select technology developers as the municipality's strongest asset in successful implementation and seeing other municipalities as not allocating adequate budgets to transform an "off-the-shelf" device to a functional assistive technology in cooperation with the technology developer. It is of interest to further research to investigate what separates the municipalities emphasizing contracts and those that emphasize collaboration as a means to successful implementation of assistive technologies.

Analysis of values created the main discussions, especially the column dignity in the matrix-produced reflections. I experienced that the informants accepted the division into utilitarian and fairness concerns easily, but the line regarding dignity was more problematic and more valuable at the same time. The problematic aspects consisted of different conceptions of dignity among the interviewees, but also among those working with assistive technologies, according to the informants. Several informants expressed that the column dignity had content that was at the core of their efforts in the health services, and likewise one technology developer said that this form of documentation of how an assistive technology might affect quality of life is central to their planning and sales as a technology firm. Dignity expressed the types of change that several of their customers sought.

When discussing the Health system as a stakeholder, one informant said that what mattered in this time of technological hype was to be frugal. Since frugality is one central concept in the recent proposal for Responsible Innovation in Health [18], I pursued this theme and asked why she used the word *frugal* and what she meant by it. She replied that one has the responsibility to ensure that patients and users receive proper care, something that cannot be left uniquely to technologists and that when spending large amounts of public money, one has the responsibility to ensure that these funds are spent well. I raised the theme if one should replace the heading welfare with the heading *frugality* in the overall Matrix, but she said that while it made sense for the stakeholder of the Health system, it did not apply well to the other stakeholders.

Regarding the case and display of information, one interviewee expressed skepticism if this GPS system was a valuable case to study as an example since GPS localization currently was not seen as

only unproblematic—contrary to the situation 5 or 10 years ago—but also highly desirable by everyone. In all the interviews, I had different discussions regarding specific interpretations of the content of the cells, but no one expressed that any of content was erroneous even though some were surprised or suspicious to the content of one or two cells. Displaying the knowledge status for an intervention with assistive technologies was conceived as valuable, both as providing the specific state of affairs and as a general approach. Several informants engaged in discussion if I had presented the right critical factors and of the internal links between the critical factors as well as their sequence and placement in the matrix.

In addition, we had discussions concerning the layout and the presentation. In general, the interviewees agreed that one should attempt at diminishing finer nuances and limit the presentation of the potential consequences as certain, uncertain, and ignorance. The weighing should likewise consist of the categories very important, important, and unimportant. I will not pursue the issues of layout and graphics further in this paper.

### 3.3. Places for Responsibility

As mentioned above, the general view was that it is too complex—at least at first sight. However, the Matrix contained elements the informants found useful, and, in addition, they mentioned concrete places for using the matrix in the working and innovation processes in the municipalities as well as potential for dialogues within the municipalities and between those implementing assistive technologies and those external to the process, such as policy-makers and firms.

Some informants expressed that one potential place in the municipal innovation chains was to employ the Matrix with health professionals or care workers in order to discuss their own experiences with existing solutions under testing or prior to deployment. One informant had recently been involved in a project where care workers filled in diaries or logs to document how a new assistive technology was used in homes. This exercise garnered an impressive amount of information regarding both the home-dwellers and the care workers interactions with the devices and with the elderly, and also regarded the organization of the services. However, what they lacked was a method that could structure what affected welfare, dignity, justice and fairness when they all discussed their individual experiences as a collective. The informants mentioned the utility of structuring discussions with care workers, however, they did this in earlier phases in order to structure concerns, thoughts and interests over novel solutions. Such dialogues are necessary and useful, but they have a tendency to be dominated by a few central themes to the detriment of less acute problems that may be of lower significance to some, but that does not mean that the themes are irrelevant. In addition, a possible use could be to investigate the relations between and experiences or views of different categories of health professionals or care workers in the home-based services.

All informants who took the perspective of using a matrix as a structure for dialogue between care workers also raised the theme of facilitation. They expressed concern that facilitating would need to be based on some specific skill set. However, this is not different from other situations where one wishes to use input from employees in developing the workplace.

A different perspective was to see the potential value of the Ethical HTA Matrix as a planning and documentation tool to prepare for the introduction of an assistive technology and structure the discussion with technology suppliers as well as mapping potential pitfalls. A similar, albeit somewhat different approach, was to apply the matrix as an intermediate mapping tool before setting out on a gain's analysis and risk management as it provided a (too large) overview over the values at stake for the relevant stakeholders. One informant said that such work was often done more or less intuitively while the risk management and the analysis of gains had a rigorous structure. In such a use, the Matrix could be applied at an early stage in order to filter and select desired effects and to concentrate on some specific gains.

Even though several informants drew a picture where politicians set unrealistic goals or goals that would lead to a near-future impasse because of obsolete technological products, they did not mention

that the Matrix held a potential to be applied at a political level or as a dialogue instrument between themselves as experts and politicians.

### 3.4. The Processes of Filling in the Matrix

As will be further developed in the next section, the systematic approach to the documentation of knowledge triggered interest for all informants except one. As mentioned above, there clearly exists sources of knowledge concerning the assistive technologies internally in companies and in municipalities to which outsiders do not have access. Questions are related to the creation of the presented matrix related to the literature searches, the amount of sources, the validation process, the workload, and the possible validity of the findings. However, some informants said that the main challenge in assistive technologies is not so much how to systematize what is known, but rather to bring the practical experiences with assistive technologies from the care workers to the decision-makers, and then to act on this knowledge in order to create improved services.

I emphasized that the content in the Matrix built substantively upon the Socratic approach [22,46, 103], and that these were the questions guiding the search for the central value topics. However, this step was not commented upon by any of the informants. Not even those who saw it as a useful way of structuring information in planning and implementation processes.

When I presented the validation phase—obtaining knowledge from the different stakeholders of how they rated the different values—there was surprisingly little reaction to the process, but as accounted for above, the results with font sizes and positive or negative impacts triggered discussions.

## 4. Discussion

### 4.1. A Tool for Decision-Makers?

The main reaction to the matrix as containing an abundance of information might depend upon my presentation of it and the fact that I did little to structure the content beyond that of what the Socratic approach provided. However, as has been mentioned by Kaiser, Millar, Thorstensen and Tomkins [24], the Matrix is not specifically user-friendly, but it is rather its structuration approach that might provide clarity. In addition, it also raises the issue of what kind of expertise is needed for applying it. Some of the informants found some parts of the Matrix easy to understand and illuminating while others found that it had little relevance or that it was difficult to grasp. It might be necessary to have some training or knowledge of applied ethics in order to perform an analysis according to the Socratic approach and systemize the findings according to a Matrix. However, as one informant said, all forms of discussion that are intended to lead to an improvement require a form of moderating that is based on skills. What remains as a challenge is that the required skills might not be well or evenly distributed.

The degree to which one can conclude whether or not the Matrix is a valuable tool for decision-makers depends on what one perceives the decision-making problem to be. Traditionally in HTAs, the problem is framed as presenting the correct information in a relevant format to decision-makers.

Garrido and colleagues describe the rationale behind HTAs as "to optimize care using the available resources" [104]. There are several possible interpretations of what such a phrase might mean since both "care" and "resources" are terms that can be described with different meanings in different settings. In a later chapter, Røttingen, et al. [105] posed relevance as a primary quality in HTAs for decision-makers and policy-makers. An overarching question then becomes how one can provide information as to the optimization of care using the available resources in the most relevant manner. Undoubtedly, relevance is a difficult criterion as well. If one stays solidly inside a bio-medical and cost-effectiveness frame of mind, it is possible to analyze a procedure on how well it reduces or enhances a certain bio-medical process and assess the cost of the procedure and the assumed economic benefits for the health delivery system in short, mid and long range. However, in such a frameset, the ethics, values and social implications seemingly disappear, and when we open up for ethics and

social dimensions of medicine, things tend to become more complicated. In a much cited paper, Porter argues for a high value for patients as the highest goal of the health delivery system, with "value defined as the health outcomes achieved per dollar spent" [106]. This definition seems blind to the different usages of value that exist in current political debate and to the use of the health system as a means of political structuration of goods whereby disadvantaged groups should receive more aid than privileged groups—as well as alternate axiologies included in that of the HTA itself [103,107,108].

There are different epistemological cultures governing the ethical and the medical where the onus placed on evidence seems to prevail [51]. May, Mort, Williams, Mair and Gask [101] presented a sociological understanding of HTAs as a form of normative evaluation that connects to the ever-increasing emphasis laid on evidence for changes in practice. However, in addition to the epistemological point, Mol [109] sees a different and more fundamental divide in the ontologies of the objects of health research and the human body. Mol contrasts between "disease", which is what is inside the body, and "illness", which is the way we talk about, value and give meaning to "disease", and posits that these two might coexist in the same space and time, just as social values, ethics and clinical effectiveness in HTAs. May, Mort, Williams, Mair and Gask [101] emphasized that HTAs are not only about evidence, but also represent an inherent thought of modernization of treatments through research and innovation.

One of the strengths identified by some of the informants, though not all, was the Matrix's ability to provide an overview of both "illness" and "disease". In addition, other respondents saw the Matrix as providing an important first step in systematizing the different values or issues and their possible effects on different stakeholders. This first step would then provide the foundation for a form of governance mechanism with which they are more accustomed, such as risk management or value realizations' tools.

### 4.2. Where in the Processes

What emerged through the interviews on the innovation processes in the municipal health sector might improve the understanding of where one might open up for mutual learning, anticipation of problems, identification of alternatives, and discussions underlying outcomes with multiple values [31]. In the literature on RRI, a view on innovation as the creation of novel technological artifacts is very often presupposed [110]. However, in the health services, an important aspect of innovation is novel ways of working. As Wouters, Weijers and Nieboer [97] among others point out, technology implementation changes the working processes and thus might also affect nurses and care workers' values. Studies of Norwegian innovation strategies for ageing at home further indicate that there is a lack of structured approaches or tools to manage and include workers in the innovation processes [111]. Creating a space for organizing discussions around central values and their conceptualizations among health professionals seems to be a very relevant place for intervening in the innovation process. According to Blok and Lemmens [55], it is highly likely that different stakeholders have different priorities or interests, and they point out that this form of reluctance to cooperation based on strategic motives is not well discussed in RRI. In providing a structure that could open up for identifying the values and interests either as a tool for dialogue or as a tool for structuring experiences, the Matrix might also fill a space central to the innovation processes in healthcare.

Reijers, Wright, Brey, Weber, Rodrigues, O'Sullivan, and Gordijn [15] separate between ex-ante methods, intra methods and ex-post methods for distinguishing between ethics or value inclusion before, during and after product launches in research and innovation. What I understood from the interviews, was that there could be a place for the Matrix in a phase where prototypes are tested for improvement together with care workers and health professionals, which is then what Reijers et al. describe as the intra phase. Using a Matrix to structure these early experiences with a novel technology corresponds to a bottom-up version of the Ethical Matrix which leaves the participants to fill in every cell in the Matrix [35]. For such uses, they signal that the process might fall victim to partisan views or that the participants misunderstand the methodology with only open cells before them. It is precisely

in order to avoid misunderstandings and/or partisanship that I applied the themes from the Socratic approach that serve to specify what the values and issues at stake might be [22,23,46]. However, none of the interviewees expressed comfort that there was such a structured approach underlying the Matrix. In addition, the informants suggesting such a use said that it could be used for subjective experiences and as a basis for further discussions. As underlined by one informant, the Matrix allowed different views to co-exist without having to neglect one issue just because a different issue was under discussion. From the perspective of RRI, encountering and discussing different sets of values could enhance reflexivity among the actors in the health innovation system [13].

If one accepts that innovation is just as much a social activity as technological invention, then the planning and selection use mentioned by several informants would qualify it as an ex-ante method in Reijers et al.'s terminology [15]. In contrast to the bottom-up approach, this usage would qualify as top-down or desk-research-based use of the Matrix [35]. Behind the thinking on early-stage use of the Matrix, there are two different rationales, according to the informants. There were those seeing it as a mapping tool to systematize the current knowledge status in the field, and there was also the approach mentioned that the Matrix could identify central values that the municipality could aim at realizing for different stakeholders through introduction of an assistive technology. In this usage, the Matrix could become a vehicle for value-based governance if it is applied together with and validated by the relevant stakeholder groups. Such an approach would seem to be in accordance with what von Schomberg [7] sees as central to RRI with the realization of public values together with economic values. As an overall mapping tool of the distribution of potential impacts, it would be useful. However, one of the basic ideas behind the concept of ethical tools as well as structured tools is that they should lead to some type of discussion or action based on the outcomes of application of the tool [112,113]. If the Matrix with its colors and fonts becomes limited to indicating what we know and what we do not know, and striving to make every cell as close to green as possible, then it is more of an epistemic tool than a normative tool. Nevertheless, there is a possibility for reflecting upon alternative ways of achieving the relevant values.

If the difference between a strong and a weak form of RRI is how strong the links are between stakeholders and policy process, as suggested by Coenen and Grunwald [27], then the use of the Matrix for inclusion of workers' views and experiences into the decision-making context would qualify as strong, while the use of the Matrix as a mapping tool independent of stakeholder input would qualify as weak.

When it comes to the discussions during the interviews regarding knowledge and the existence of a range of unpublished local municipal studies or consultancy reports, there are some concerns. As the use of assistive technologies tends to understand life conditions within a biomedical rather than social frame [109], there seems to be a peril of conducting too-narrowly defined studies. In addition, implementation of assistive technologies depends on local features and the transfer from one context to another could be problematic. Obviously, there is a risk for ignoring unpublished studies when making knowledge reviews. Many technology developers are also reluctant to put their studies into the public domain while sharing these with decision-makers as they are aiming for patents or other forms of intellectual property rights. This situation makes it highly difficult to conduct independent audits or assessments of these solutions.

### 4.3. Experiences with the GPS System as Case

I have struggled with the reasons why the respondents did not discuss the proposed case of the GPS system more and the different findings from the literature. One obvious element is that I intended to discuss the methodology and not the case specifically since it is the Matrix that is the object for research. Furthermore, as one informant said, GPS localization is not an issue any longer in the public debate as was the case up until 5 years ago. Two of the informants would qualify as "issue advocates" in taking a very firm stance that safety for users was the central social and political question and GPS systems could and would deliver such safety (see Pielke [114]). However, several

of the informants wished to know what was behind the red colors, i.e., what is not known or are just hypotheses concerning GPS localization systems. This type of epistemic use of the Matrix points toward the possible uses for other tools or instruments that might provide insights into uncertainties and risks. As narrated above, several informants raised very specific points on which they disagreed or did not understand properly. By using these questions and queries as indicators, the proposal by Fitzgerald and Adam [9] to introduce forms of decision-support systems as a means to enhancing responsibility in planning, procurement and implementation of assistive technologies seems promising since it could contribute to steering decision-making towards addressing good aging at home with a higher epistemic quality.

### 4.4. A Brick in the Bridge

Demers-Payette, Lehoux and Daudelin [13] point to the different logics in health and industry where the former is stability seeking and the latter is risk- or gain-seeking. They recommend to address the differences in the value systems and the social practices in the health care system and the innovation chains where health care is strongly resistant to change with a focus on medical needs while industry turns around rapidly with a focus on lucrative opportunities. The expression of frugality as an ideal would qualify as typical of the health care system in this respect, while the view on the business case could illustrate the industry. Accordingly, there seems to be a gulf between these logics that could be bridged. The question is then to what extent the Matrix could be successful in contributing to making such a bridge. As mentioned above, respondents identify the Matrix as a tool for intra-stakeholder deliberation. A different question is how it could work as an inter-stakeholder tool for addressing value differences. The Matrix has already been applied in inter-stakeholder workshops [24,34]. What was most indicative of the potential for inter-stakeholder utility, I believe, is the amount of time and interest the respondents paid to the different stakeholder groups and if they saw the other groups as relevant. In the interviews, most respondents acknowledged the other stakeholders as relevant, and the representatives from the municipalities expressed concern over stifling innovation through an increasingly hierarchical market with a few central providers using the smaller firms as subcontractors. On the other hand, the technology developers expressed concern over instances of too strong-handed use of procurement power. These two latter instances could well be understood as conflicts based on differences in power that supersede the analysis by Demers-Payette, Lehoux and Daudelin [13], who investigated internal logics as well as Chatfield, et al. [115] who described an industry very eager to cooperate with the stakeholders. According to Blok and Lemmens [55], such differences in power are inherent to innovation processes and "[i]t is presumable that power imbalances are especially at stake in the case of grand challenges, exactly because of the different problem definitions and different value frames of the stakeholders involved" [55]. The challenge is consequently to find some domain where the parties might become responsive to each other. Here, it is noteworthy that some informants saw dignity as an essential category to which they seemed to attach large significance. Dignity constitutes both a central business case for suppliers and developers of assistive technologies and a central mandate for the health care system. However, as Sontag [116] accentuates, our perceptions of pain and dignity are phenomena determined by their surrounding narratives and frames. This background dependency would then presuppose a discussion of the frames, visions and rationales behind engaging in and showing concern for dignity rather than approaching dignity as a discrete entity that might be addressed directly. A central ambition within RRI is to discuss such framing effects. However, these effects seem difficult to address through standardized tools such as the Matrix alone (see Zwart, Landeweerd and Rooij [93]). In my combined approach here, one could investigate such framing effects through an increased attention to the theme "Is the symbolic value of the technology of any moral relevance?" raised in the Socratic approach (see Appendix A) [22].

*4.5. Methodological Considerations and Limitations*

The main limitation in this study is the number of interviewees and their selection. Thirteen respondents might not provide an exhaustive presentation of the possibilities for intervening in health innovation processes nor for the possible uses of the current Matrix. In addition, I selected respondents from the area surrounding the country's capital, which has a much higher population density than other parts of Norway, but then again it is lower than many other parts of the world. Furthermore, no respondents expressed concerns over lack of funding for assistive technologies—a situation I could imagine might be different in other places. These considerations might affect the transferability of the findings.

Since I approached the interviewed experts in a co-constructivist mode with open questions and some overarching themes to discuss, with the assessment methodology and its possible uses at the center of attention, other aspects suffered from lack of time and attention (see above). I could have used a different approach with a clearer focus on the case, but that would affected the attention to the methodology. In addition, the literature searches underlying the Matrix and its completion were based on GPS systems in general and some GPS systems with next of kin as first responders. I want to signal that there is no such thing as a generic GPS system, but a range of different solutions with their own composition and designs implemented in individual health care systems with unique organizational features. Furthermore, the case and validation process for the case was limited to one town and within only one health care system. This limited scope might have affected the validated Matrix (Appendix B) and consequently the informants' view on the content of the Matrix. A more thorough validation process would have placed additional strain on health professionals and care workers, as well as on persons with dementia, and I decided not to burden already strained or fragile persons.

## 5. Conclusions

Seen from the perspective of RRI, the current contribution finds that there is support among central stakeholders in assistive technologies for structured approaches that might secure the realization of economic and public values. Explorations into the configuration of responsibility in innovation process are still in their infancy, and this is particularly so in the public sector where the conditions for innovation are radically different from the private sector. The main differences are on the one side that citizens cannot just opt for a different country or a different municipality as they can with respect to the acquisition of private goods and services, and on the other side that authorities are legally mandated to provide some goods or services to its citizens regardless of their ability to pay, give feedback or even desire these services. Nevertheless, innovation also takes place in the public sector, and in this contribution, I have shown that there seems to be a place for responsibility in the innovation pathways in the health sector. The main place for normative considerations and reflexivity is in the interaction with health professionals and care workers during the implementation of novel assistive technologies. A strength of the Matrix in this respect is that it might pay attention to both the physiological and socio-cultural aspects of disease or illness simultaneously, whereas more epistemic dimensions of responsibility and knowledge or uncertainty management concerning the affected values might in addition benefit from more systematized approaches, such as a Matrix.

An additional factor for RRI could be to study the effect of different concepts in the health care system. For example, the introduction of *frugality* as an ideal could well open up novel avenues for understanding how decision-makers understand their mandate and how private values affect public actions.

Even though the integration of the Socratic approach into the Matrix did not receive any response from the informants, this feature remediated to some extent the concern expressed by Mepham, Kaiser, Thorstensen, Tomkins and Millar [35] over partisanship and misunderstanding since it directs the users to what are the relevant concerns.

Concerning the ability of the Matrix to build bridges between health and industry, I remain a bit skeptical due to the power struggles that seem to be present. If these power aspects could be addressed

and the discussion would revolve around the different rationales or frames for intervening in dementia care, one might be somewhat more optimistic. However, if this transformation first takes place, there would be little need for a tool such as the Matrix because then the different parties would already have acknowledged each other's values.

Some changes are however necessary to apply a structure such as the Matrix in the innovation systems in assistive technologies. When it comes to the display of information, some findings are relevant such as a limited color range and a limited number of facts and values. The issue of critical factors is specifically a central one in assistive technologies due to the large room for maneuvering for local authorities in adapting service structures to these solutions. On a substantive level, the requirements for introducing such structured frameworks would demand several experiments at the municipal level in terms of opening up deliberative structures together with the care workers and health professionals—as well as challenging existing cognitive, epistemic and normative divisions of labor between these and the workers and the employer.

Testing such a modified approach then remains. In addition, in testing approaches such as the Matrix in healthcare, one should proceed with some caution for (at least) two reasons. The first reason is that this is a system under constant pressure both to deliver quality services and to adapt to the changing nature of service delivery. Consequently, one should refrain from placing burdens on the services and the personnel. The second reason for caution is that even though the Matrix might function as a transparent tool concerning the steps, it follows the outcomes of a matrix-process into the decision-making process, which also needs to be clear and transparent. Without structures in place for such an outcome transparency (see e.g., Rowe and Frewer [100]), the whole process (and the products) might lose legitimacy.

**Funding:** The project: 'The Assisted Living Project: Responsible innovations for dignified lives at home for persons with mild cognitive impairment or dementia', is financed by the Research Council of Norway under the SAMANSVAR strand (247620/O70). Collection and storage of data is granted under Norwegian Centre for Research Data No. 47996 and by written consent by all participants providing personal information.

**Acknowledgments:** I am very grateful to all the persons who have dedicated time to the Assisted Living project, and I thank the reviewers for valuable improvements and suggestions.

**Conflicts of Interest:** The author declares no conflict of interest.

## Appendix A. The Ethical HTA Matrix

| | Well-Being | Dignity | Fairness |
|---|---|---|---|
| Primary user | What are the effects according to purpose?<br>What resources are needed?<br>Q1 What is the severity of the condition to be addressed? May this change?<br>Q8 What other benefits or harms are there to the primary user? Please consider the implementation, use or withdrawal of the technology<br>Q4 Does the technology involve disease prediction? How are false test results, overdiagnosis, futile or harmful treatment addressed? | Q15 Is the symbolic value of the technology of any moral relevance for the primary user? (Prestige, status?)<br>Q16 Are there moral challenges related to components of a technology for the primary user?<br>Q17 Are there any related technologies that have turned out to be morally challenging with respect to the direct user?<br>Q12 Does the technology in any way challenge or change the relationship between patients and health care professionals or between health professionals?<br>Q10 Will there be a moral obligation related to the implementation, use, or withdrawal use of a technology? (e.g., consent)<br>Q6 Does the technology challenge a user's values or social relations—or might it affect a user's religious convictions?<br>Q5 Does the implementation, use, or withdrawal of the technology challenge a user's autonomy, integrity, privacy, dignity or interfere with basic human rights?<br>Q3 Might the widespread use of this technology change user's social roles? (Does it change the prestige or status, the conceptions, prejudice or status of persons with certain characteristics [e.g., old age]?) | Q9 Can the implementation, use, or withdrawal of the technology in any way conflict with existing law or regulations or pose a need for altered legislation?<br>Q7 How does the implementation, use, or withdrawal of the technology affect the distribution of health care regarding the users? (Justice in allocation, access, and distribution).<br>Q2 What patient group is the beneficiary of the technology? (Are they particularly vulnerable, have low socioeconomic status or priority, or are they subject to prejudice? |
| Health professionals (or care workers) | What are the effects according to purpose?<br>What are the resources needed?<br>Q8 What other benefits or harms are there to the health professionals? Please consider the implementation, use or withdrawal of the technology. | Q20 How does the technology contribute to or challenge or alter health professional's autonomy?<br>Q15 Is the symbolic value of the technology of any moral relevance for health professionals?<br>Q12 Does the technology in any way challenge or change the relationship between patients and health care professionals or between health professionals?<br>Q6 Does the technology challenge health professionals' social or cultural values, institutions, or arrangements or does it affect their religious convictions?<br>Q3 Does the widespread use of this technology change the role of health professionals? (Does it change the prestige or status of the disease, the conceptions, prejudice or status of persons with certain diseases?) | Q9 Can the implementation, use, or withdrawal of the technology in any way conflict with existing law or regulations or pose a need for altered legislation?<br>Q2 What professional group will work with the technology? (Are they particularly vulnerable, have low socioeconomic status or priority, or are they subject to prejudice?) |

| | Well-Being | Dignity | Fairness |
|---|---|---|---|
| Health delivery system | What are the effects according to purpose?<br>What are the resources needed?<br>Q8 What other benefits or harms are there to the health delivery system? Please consider the implementation, use or withdrawal of the technology<br>Frugality: Does the technology deliver greater value to more people using fewer resources? Does the technology presuppose a larger technological infrastructure? | Q11 How does the assessed technology relate to more general challenges of modern medicine? (Underdiagnosis, undertreatment, medicalization, overdiagnosis, overtreatment, reduced trust)<br>Q20 How does the technology contribute to or challenge or alter health professional's autonomy? | Q9 Can the implementation, use, or withdrawal of the technology in any way conflict with existing law or regulations or pose a need for altered legislation?<br>Q7 How does the implementation, use, or withdrawal of the technology affect the distribution of health care? (Justice in allocation, access, and distribution). |
| Technology providers | ?<br>Q8 What other benefits or harms are there to the technology providers? Please consider the implementation, use or withdrawal of the technology | ? | Q21 What are the interests of the producers of technology (industry, universities)? |
| Next-of-kin | What are the effects according to purpose?<br>What are the resources needed?<br>Q8 What other benefits or harms are there to the next-of-kin? Please consider the implementation, use or withdrawal of the technology | Q12 Does the technology in any way challenge or change the relationship between users and next-of-kin or between next-of-kin? | Q7 How does the implementation, use, or withdrawal of the technology affect the distribution of health care? (Justice in allocation, access, and distribution). |
| Society as a whole | What are the effects according to purpose?<br>What are the resources needed?<br>Q8 What other benefits or harms are there to society as a whole? Please consider the implementation, use or withdrawal of the technology. | Q16 Are there moral challenges related to components of a technology that are relevant to the technology as such?<br>Business model: Does the organisation that produces the innovation seek to provide more value to users, purchasers and society? | Q7 How does the implementation, use, or withdrawal of the technology affect the distribution of health care? (Justice in allocation, access, and distribution). |
| Climate | Decrease of greenhouse gas emissions through product lifecycle;<br>Increase of greenhouse gas sinks | | |
| Ecology | No parts of the product lifecycle cause unnecessary harm to the environment;<br>A maximum of ecosystems to be protected through product lifecycle | Product lifecycle limits harm to nature to a minimum | No ecosystems suffer disproportionally more than others |
| Other stakeholders | | | |

The Ethical HTA Matrix with structure from Mepham, Kaiser, Thorstensen, Tomkins, and Millar [36] and content decided by Hofmann, Droste, Oortwijn, Cleemput, and Sacchini [22], andHofmann [23] and supplemented with aspects from Responsible Innovation in Health Pacifico Silva, Lehoux, Miller, and Denis [18].

## Appendix B. The Validated Matrix

| Primary user | Welfare | | Dignity | | Justice | |
|---|---|---|---|---|---|---|
| Critical factors: Studies indicate that an average person with dementia might live at home up to one year though GPS | Live at home (+) | Some users might reside at home for a longer time | Able to ask for assistance (+) | Most find it easier to request assistance | Everyone with the same needs get proportionally equal access to the same services (?) | Without family living nearby, there is a need for public solutions |
| Critical factors User ability to benefit from GPS Abilities and habits: understanding traffic; going for walks Family living nearby | Mail, shopping, waste disposal (+) | Most experience increased mastery of daily tasks | Trusting the services (?) | It is uncertain how GPS increases trust in the services | Consent to use (+) | Ability to consent GPS tracking legally sanctioned |
| | Outdoor movement (+) | Most get around more | Decide what activities to partake in (+) | Several seem to partake in more activities | | |
| Social connections nearby Understanding the design of the GPS tracker Ability to consent | Going for walks (+) | Those with the habit report more walking | Decide on the service measure (+) | GPS is a service measure where consent is central | | |
| Personal convictions<br><br>Functioning GPS system | Vacations (+) | It seems possible for more people to go on holiday | Decide where to go (+) | Increased opportunities for all to decide on where to go | | |
| GPS accuracy and updating frequency | Be found (+) | Users are located and found | Contact with family (+) | Increased contact with family | | |
| Organisation of health services Solutions for those without family living nearby | Affordable services (?) Experience that the service is worth the cost (+) | Many experience the service as good, but the quality seems variable | Contact with friends (+) | Most can maintain contact with friends | | |

| Next of kin | Welfare | | Dignity | | Justice | |
|---|---|---|---|---|---|---|
| Critical factors: User's ability to benefit from GPS<br><br>Well-functioning GPS system GPS accuracy and updating frequency<br><br>Adequate training | Own safety (+) | Everyone experience increased safety | Safety for next of kin (+) | Everyone experiences increased safety | Adequate sharing of care burden (?) | Most experience less relief but little is known about the fairness of the arrangement |
| | Relief in caring (÷) | Most experience less relief | Freedom for next of kin (+) | Everyone experiences increased freedom | | |
| Solutions for those without family living nearbyOrganization of health services | Own job / career (+) | Some find more time for work / career | Freedom (+) | Most experience more freedom | | |
| | Time to remaining family (?) | Uncertain how many find time for remaining family | Peace of mind (+) | Peace of mind is the largest effect | | |
| | Time to maintenance of GPS equipment (÷) | Several use time for maintenance and charging | Role changes in family (?) | Next of kin become carers–uncertain if it is negative or positive; it is a change | | |
| | | | Knowing where next of kin is (+) | One has always the possibility to track | | |
| | | | Understanding the services (+) | Most seem to understand the services better | | |
| | | | Understanding the technology (+) | Most seem to understand the technology better | | |

| Employees | Welfare | | Dignity | | Justice | |
|---|---|---|---|---|---|---|
| Critical factors: Employees Critical factors | Feeling safe at work (+) | Most feel safer at work | Freedom to provide healthcare (+) | Most experience increased time for healthcare | Users understand the legal grounds for the service (+) | Seems to be increased attention to consent with GPS |
| Well-functioning GPS system Configuration of GPS system to electronic patient register | Understanding the seriousness of alarms (÷) | With less knowledge of the user, the seriousness of alarms might become difficult to estimate | Understanding the technology (?) | Uncertain if the health workers increase their understanding | Next of kin understand the legal grounds for the service (?) | Uncertain whether next of kin become more informed about the legal aspects |
| Quality and structure of the control panel for the GPS system | Correct user location information (+) | Everyone can get a precise location | Adequate training in technology (?) | Uncertain what kind of training that is given | Next of kin understand the privacy regulations (?) | Uncertain if next of kin understand or maintain privacy rules |
| Organization of health services Financing of health services | Understandable technical infrastructure (?) | Uncertain if the infrastructure is understood | Recognising users' ability to benefit from the technology (?) | Very uncertain what happens with the match between user and solution | Agreement between service and next of kin on measure (÷) | Uncertain, but clear potentials for disagreements over the service |
| Ability to consent User contact | Competent colleagues (÷) | Can become difficult to find adequate personnel | Knowing the users' cognitive condition (÷) | There is a danger for lower understanding of users (if less contact) | Knowing who is responsible for control of the GPS (+) | Difficult to estimate, but work sharing seems to become clearer |
| Increased research Personal convictions regarding health services Adequate training | Vulnerability to technological errors (÷) | Increased vulnerability to errors Wrong location might be critical | Knowing the users' general condition (÷) | There is a danger for lower understanding of users (if less contact) | Routines for who is responsible for locating users (+) | The routines for responsibility sharing become clear |
| | Vulnerability to inherent limitations in technology (÷) | Increased vulnerability if GSM/GPS does not work indoors | Having adequate info on the user in case of alarms (÷) | Novel challenges seem to arise when workers know where a user is but not what s/he does | Routines for vacation and travels (÷) | It seems that changing routines might be a challenge |
| | Resources for rescue (+) | Less time is spent on rescues | More time to provide healthcare (+) | Most experience more time to healthcare | | |
| | Next of kin assisting in finding users (+) | Next of kin will find users in most cases | Conflicts with users (+) | Less conflicts arise when users can move freely | | |
| | No. of alarms (÷) | More devices with alarms = more alarms | Confidence with modernization of healthcare (+) | Work changes character with more attention to maintenance of devices | | |
| | No. of false alarms (÷) | Most alarms are false alarms | | | | |
| | Relaying on next of kin's knowledge (?) | Difficult to assess next of kin's knowledge | | | | |
| | Adapt services to user (+) | Valuable tool for personal adaptation in most cases | | | | |
| | Next of kin responding to alarms (+) | Next of kin participate to a large extent | | | | |
| | Next of kin locating users (+) | Next of kin participate to a large extent | | | | |
| | Cooperation with next of kin (?) | Next of kin participate to a large extent, but unclear if it improves to cooperation | | | | |
| | Early intervention (+) | Early intervention is assumed to have positive effects | | | | |

| Health system | Welfare | | Dignity | | Justice | |
|---|---|---|---|---|---|---|
| Critical factors: Organisation of health services Quality and struc-ture of the control panel for the GPS system | Efficient organization (?) | Not clear how GPS affects organization | Increased user quality of life (+) | Most users report increased QoL | Equal access to services for all citizens regardless of social situation (?) | Without family living nearby, there is a need for public solutions |
| Pricing mechanism with tech supplier Adequate training Early interventions Functioning GPS system Solutions for those without family living nearby | Efficient service (+) | Services seem to become more efficient if differentiated | Postponed need for adapted home (+) | A minority of users seems to reside at home for some time | User consent (+) | Increased attention to consent through tracking technologies |
| User ability to bene-fit from GPS User's abilities and habits | Efficient cooperation in the service (?) | The service cooperation is unclear | Postponed need for assistive living facility (+) | A minority of users seem to have postponed use for different housing | | |
| | Robust technical infrastructure( | Vulnerability increases through multiple systems, but SafeMate Pro seems robust | Active users (+) | Most users become more active, but efforts are needed for those without the habit | | |
| | Easy upscaling of users (+) | The system seems flexible when it comes to upscaling, but little research | Less practical assistance (+) | Flere klarere hverdagsgjøremål, men er avhengige av tjenester for å opprettholde aktivitet | | |
| | Economic savings (+) | Uncertain, but there are indications of savings–related to home residency | Longer user residency at home (+) | A minority seems to be able to reside at home for some time | | |
| | Affordable upscaling of users (÷) | Uncertain, but there does not seem to be any savings related to upscaling as such | Early intervention (+) | Early intervention is expected to increase positive effects | | |
| | Staff reduction (÷?) | Very unsure, but little indicates fewer employees | Finding persons faster (+) | Users are located faster | | |
| | Certainty regarding expenses (?) | Very uncertain, and especially due to rapid technological changes | Quality control of services (?) | GPS increases the need for the control of services. Little research on the quality control. | | |
| | Certainty regarding future savings (?) | Very uncertain, also affected by social and technological changes | Creating new services (+) | Re-organisation of health services seems to be a recurring consequence | | |
| | User payment for services (?) | Very uncertain how user payment is decided; local variations | Coordinating services with technology (?) | GPS increases the need for the coordination of services, but little research on the actual practice, | | |
| | | | Employees mastering technologies (?) | GPS demands increased tech mastery, but unclear if mastery takes place | | |
| | | | Cooperation with next of kin (+?) | Clearly increased cooperation, but there remains challenges of coordination and quality control. | | |

| Tech supplier | Welfare | | Dignity | | Justice | |
|---|---|---|---|---|---|---|
| Critical factors Procurement policies | Assured income (?) | Uncertain, but most agreements are over 3 years or longer | Recognition for suppling important solutions (?) | Uncertain, depends on the dialogue with procurer | Competitive rules independent of products (?) | Uncertain as many municipalities already have specific systems |
| | Avoid larger uncertainties (?) | Very uncertain, but the terms of agreement tend to be clear–however novel challenges often arise | Proximity to users (?) | Uncertain, depends on follow up by both parties | Fair use of procurement power (?) | Uncertain, large procurer might set demanding terms |
| | Avoid long-term expenses (?) | Very uncertain since it depends on both agreements for development and unforeseen events | Proximity to procurers (?) | Uncertain, depends–in addition to business culture–on personal factors | Fair contracts (?) | Uncertain, but long-term agreements would indicate some fairness |
| | | | Call for tender expressed in terms of desired functions and not solutions (?) | Uncertain, presumes that procurer leaves a large amount of discretion to the tech supplier | | |

| Climate | Welfare/dignity/justice | |
|---|---|---|
| Critical factors Life cycle analysis Procurement policies | Lower emission of greenhouse gases (÷?) | Fairly certain increased emissions through plastic material Uncertain, but could reasonably lead to less driving Very likely less search operations Very likely increased usage of electric power |
| | Increased uptake of greenhouse gases(÷) | Fairley certain that there will not be any significant uptake of greenhouse gases |

| Ecosystems | Welfare | | Dignity | | Justice | |
|---|---|---|---|---|---|---|
| Critical factors Life cycle analysis Procurement policies | Avoid unnecessary harms to ecosystems (÷?) | Uncertain what recycling arrangements there are Spread of heavy metals | Limit harms to the environment to a minimum (÷?) | Reasonably certain that the heavy metals in phones and tracking devices will increase pollution | No ecosystems to suffer disproportionally to others (÷) | Reasonably certain that places where central metals are mined will suffer more than others |
| | Protect as many ecosystems as possible (?) | Uncertain how the pollution is distributed | | | | |

| Certainty of effect | Hypothesis | <25 % certain | Ca 50 % certain | >75 % certain | Broad consensus |
|---|---|---|---|---|---|
|  | 236, 112, 99 | 230, 126, 34 | 247, 220, 111 | 130, 224, 170 | 82, 190, 128 |
| Large variation in validation | 236, 112, 99 | 230, 126, 34 | 247, 220, 111 | 130, 224, 170 | 82, 190, 128 |

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
