# Peer review of "Stakeholders’ Views on Responsible Assessments of Assistive Technologies through an Ethical HTA Matrix"

_societies, doi:10.3390/soc9030051_

Round 1

Reviewer 1 Report

This is a very important topic. However, I believe that the submitted paper currently tries to answer too many different questions.

I think that there are several papers hidden in the text:

Paper 1: Results of the literature review which was performed to fill the Matrix

Currently, a lot of information is included which gives an overview of the current literature on GPS-Systems. This could be a paper in itself.

Paper 2: Feedback on the method of the Matrix

If the main focus of the paper is the method of the Matrix (which, as I understood it, is what the authors want to focus on), a very brief description of the filled in Matrix is sufficient. The details are not necessary to understand an analysis of the stakeholder’s views. In this case, paper 1 could be written first, so that paper 2 can point to paper 1 for details of the content of the Matrix.

I think that there is more in the data than the authors make of it at the moment. My impression is that reworking the presentation style and looking a bit more for abstraction (generalizations) could help. Currently, the presentation of the results is often a retelling of the interview process à la this was said by K1 and this was said by K2, this was the answer when I had asked that. I find it not so important whether something was said by K1 or K2. In contrast, it seems very interesting to me that e.g. the stakeholders representing municipalities expressed two different points of view. And I would sometimes like to know more about how the authors theorize these differences.  E.g. what could be the reason behind the fact that the technology developers are more interested in the other stakeholders’ views than vice versa?

It could also help to think about the possible audiences for the paper. Who would be interested in the results and how much generalization or detail would be helpful for them?

Paper 3: How to design a good Matrix

Here I am not really convinced of the methodology which is applied. It seems that the authors want to evaluate the graphic design (font sizes, colours) of the Matrix. They want to know which amendments of the design would be preferred by the stakeholders. The question is tackled by analysing a discussion with stakeholders including an analysis of the way people gesture towards the object of discussion. It would be interesting if there were video recordings or if the gestures were noted by hand.  Moreover, primarily analysing the reception of the design would need more engagement with the respective literature.

I would argue that the Matrix should be primarily effective in presenting information. If the topic of the paper was the effectiveness of the presentation of the information with the help of the Matrix, more engagement with experts or literature from graphic design would be helpful.

Author Response

I have combined the two replies as they were to a very large degree overlapping. Please contact me if you wish separate replies.

Reviewer 2 Report

Within the framework of RRI two methods are combined and tested for their applicability. A variant of an ethical matrix and the Socratic Method for assessment of health technologies are combined and used for assessment of a GPS orientation system in elderly care (patients with dementia). The implementation of the proposed method(s) is presented and discussed in detail. Strength and weaknesses of the conducted approach are summarized, including Stakeholder feedback, and result in a ‘skeptical’ conclusion about the impact of the proposed method for RRI studies.

The task set out by RRI is difficult to achieve. Combining ethical theory with the ever changing and progressing areas of innovation and research without hindering one or the other, is especially demanding on the communicative tools and methods of assessment available. The paper succeeds in presenting this difficulty. The topic of elderly care and tracking-devices is relevant and well chosen for research on RRI assessment.

The chosen methods are established in health research. The area of tracking systems in health services is an especially fast moving and innovative sector where RRI is dearly needed. While the structure and presentation of the study confirms to current standards, the first person narrative leads to colloquialisms. Overall the study is relevant, the approach feasible and the results are of some importance to improving RRI methods.

For the limited scope of the study – the author remarks “there is no such thing as a generic GPS system” (line 897) – the paper seems overly detailed. 

The results drawn from the study remain focused on the tool itself (“the color range is too large and three colors suffice”- line 937) and hardly scratch ways of improving communication or considering a more integrated framework – for example opening the tool-rework process not only for input, but also for ways of turning stakeholders into co-creators of an RRI framework. While the overall aim of the study is achieved, the benefit for stakeholders remains, as the author notes (line 930-931), vague.

The presentation of the chosen object of research can be improved upon. The functions and scope of the tracking-device are not well explained when introduced (line 235). Therefore it remains unclear what kind of restrictions a tracking-device can help overcome, in order to offer “freedom of movement” (line 239-240). Likewise, the privacy questions at hand cannot by judged by a reader without information of the GPS system’s functioning. This leads to an ethical issue demanding further regard: the author states “users don’t feel surveyed” (line 281-283) – yet knowing to be observed is enough to change people’s behavior. It has an impact on daily lives. Calling the topic of privacy “difficult” (line 279) cannot suffice for a study dedicated to improving RRI methodologies, but warrants a more in depth discussion of how patients are able to control personal data (reveal or conceal) when using the tracking-device.

The results (lines 520-725) are overly detailed at times and can be abbreviated once the relevant appendix B is in order. Appendix B is in need of major work and re-formatting. The table on page 28 does not fit on the page. Text size varies throughout the table. The symbols +” and “÷” are hard to discern at the best of times. The color scheme creates a misleading rainbow effect and does not clarify the given information. Descriptions and content of the table do not align.

An Appendix 3 (line 639) is missing.

Some minor notes:

Line 123: “Matrix is a based” -> “Matrix is based”

Line 125: “selected for address” -> “selected to address

Line 135: The matrix is applied to the object. Not the other way around.

Line: 159-160: The numbers and values presented have not been introduced yet. It is unclear what they represent.

Line: 170 “is not what” -> “is not: what”

Line: 172: “decisions, the validation” -> “decisions. The validation”

Line 176-179: colloquial language

Line 206: colloquial language

Line 249: “some uncertainties” – The following listed findings are not uncertain.

Line 264: colloquial

Line 265-266: “needs to be tailored” – tailored to ... what?

Line 334: “An increased in uncertainty” -> “An increased uncertainty”

Line 736: “said, in all” -> “said, all”

Line 737: “improvement needs a form of”-> “improvement, need/require/demand a form of”

Author Response

(The authors gave the same response as above.)

Round 2

Reviewer 1 Report

The presentation of the data analysis from the interviews has improved considerably, congratulations!

ad section 3.2.

However, section 3.2. needs more redrafting. The link between the results and the original starting point of the paper has to be strengthened to make the argument of the paper more convincing: E.g. returning in the results section explicitly to the 4 abilities by Wickson and Forsberg and stating in detail whether and how they are met could provide this link. This would be a good base for the subsequent discussion section.

If you decide to redraft the section along these suggestions, it is worth paying attention to the different kinds of results in section 3.2.  - instances (type 1) where the stakeholders evaluate the Matrix (e.g. accepting the content of the Matrix as truthful line 459-461; approving of the presentation e.g. more less detailed & layout; evaluating the decision by the author to use critical factors  517 ff.) and – instances (type 2) where the usefulness of the Matrix is demonstrated through its ability to help us understand/observe/reflect on key issues (views regarding needed resources to use the Matrix line 507-516; the additional knowledge which was created/documented through the use of the Matrix in the interviews 461-467; 482-506, 534 ff.). At the moment, both types of results are mixed and not distinguished. Both can be argued to be important for the four abilities by Wickson and Forsberg. Type 1 results are probably more relevant for ability 2 (there needs to be some acceptance of the tool for it to engage a range of stakeholders for the purposes of mutual learning). But it seems to me that the rich data analysis included in this section relating to type 2 results is even more important for a study using Wickson and Forsberg.

Author Response

I very much agree with these comments. In addition, I think that they require some rewriting in the discussion section. Please see these changes as well.

I have now restructured 3.2. in line with the smart suggestions (reflection on problems, mutual learning and value analysis), but additionally given some more flesh to the bones in 4.2. regarding the RRI relevance of the findings.

Thank you!!!!